# Transport and Permeation Properties of Dapivirine: Understanding Potential Drug-Drug Interactions

**DOI:** 10.3390/pharmaceutics14091948

**Published:** 2022-09-14

**Authors:** Ruohui Zheng, Guru R. Valicherla, Junmei Zhang, Jeremy Nuttall, Peter Silvera, Leslie J. Marshall, Philip E. Empey, Lisa C. Rohan

**Affiliations:** 1Department of Pharmaceutical Sciences, School of Pharmacy, University of Pittsburgh, Pittsburgh, PA 15213, USA; 2Magee-Womens Research Institute, Pittsburgh, PA 15213, USA; 3International Partnership for Microbicides, Silver Spring, MD 20910, USA; 4Advanced Bioscience Laboratories, Rockville, MD 20850, USA; 5Preclinical Microbicide and Prevention Research Branch, Prevention Sciences Program, Division of AIDS, National Institute of Allergy and Infectious Disease, National Institutes of Health, Bethesda, MD 20892, USA; 6Department of Pharmacy & Therapeutics, School of Pharmacy, University of Pittsburgh, Pittsburgh, PA 15213, USA; 7Department of Obstetrics, Gynecology, Reproductive Sciences, School of Medicine, University of Pittsburgh, Pittsburgh, PA 15213, USA

**Keywords:** dapivirine, drug transporters, tissue permeability, drug-drug interactions, HIV

## Abstract

The dapivirine (DPV) vaginal ring was developed by the nonprofit International Partnership for Microbicides (IPM) for reducing the risk of HIV infection. A clinical study (IPM 028) showed that concomitant use of the DPV ring and miconazole (MIC) altered DPV pharmacokinetic profile. In this work, we investigated whether or not DPV transport and permeation contributed to the observed DPV-MIC interaction. Our study evaluated the interaction between DPV and several transporters that are highly expressed in the human female reproductive tract, including MRP1, MRP4, P-gp, BCRP, and ENT1, using vesicular and cellular systems. We also evaluated the impact of DPV/MIC on cellular tight junctions by monitoring transepithelial electrical resistance with the Ussing chamber. Lastly, we evaluated the effect of MIC on DPV permeability across human cervical tissue. Our findings showed that DPV was not a substrate of MRP1, MRP4, P-gp, BCRP, or ENT1 transporters. Additionally, DPV did not inhibit the activity of these transporters. DPV, MIC, and their combination also did not disrupt cellular tight junctions. MIC did not affect DPV tissue permeability but significantly reduced DPV tissue levels. Therefore, our results suggest that the DPV-MIC interaction is not due to these five transporters, altered tight junction integrity, or altered tissue permeability.

## 1. Introduction

Human immunodeficiency virus (HIV) substantially impairs the human immune system, leading to the development of acquired immunodeficiency syndrome (AIDS). Globally, around 37.7 million people are living with HIV and about 13% of them are unaware of their infection [1]. Therefore, prevention strategies against HIV infection are crucial for ending the HIV epidemic. Dapivirine (DPV) is a non-nucleoside reverse transcriptase inhibitor, which has been widely investigated for administration via the vaginal route in the dosage forms of gel, film, and ring [2,3,4,5,6,7]. A monthly vaginal ring containing DPV, developed by the nonprofit International Partnership for Microbicides (IPM), has shown an excellent safety profile, high user acceptance, and promising efficacy in several clinical trials. In IPM 015, 97% of participants reported the DPV ring to be comfortable [3]. Two phase III clinical trials, the Ring study (IPM 027) and the ASPIRE study (MTN-020), found that the DPV ring was well-tolerated and effective in reducing women’s HIV risk. Compared to the placebo groups, the HIV-1 incidence rate among DPV ring groups per 100 person-years was reduced by 35% in the Ring study and 27% in the ASPIRE study [4,8]. In addition, in the ASPIRE study, only 18% women reported that their male partners were unsupportive for using the DPV ring [9]. Importantly, recent data from open label extension studies (DREAM (IPM 032) and HOPE (MTN-025)) for the DPV ring demonstrated improved adherence and suggested based on modelling data a greater reduction (~50%) in HIV risk [10]. The DPV vaginal ring has received a positive opinion from the European Medicines Agency (EMA) for HIV prevention in women, as well as prequalification and a recommendation from the World Health Organization (WHO) as an additional HIV prevention option for women at substantial risk of HIV infection [11,12]. Importantly, the DPV ring has been approved in several eastern and southern African countries, including Zimbabwe and South Africa, and is currently under regulatory review in other countries in the region [13,14].

In addition to HIV prevention products, the vagina is a route of administration for other products such as the treatments for vulvovaginal candidiasis. Vulvovaginal candidiasis is the second most common type of vaginal infection and accounts for 1.4 million outpatient visits annually in the U.S. [15,16]. Vaginal suppositories containing miconazole nitrate (MIC) are used to treat vulvovaginal candidiasis and superinfection due to Gram-positive bacteria [17,18]. An open label study by IPM showed that concomitant use of a single-dose MIC suppository (1200 mg MIC) with the DPV ring increased the systemic exposure to both drugs (IPM 028) [19]. Understanding the underlying mechanisms for this observed alteration in pharmacokinetic profile with concomitant use of DPV and MIC will be useful to predict potential interactions between DPV and other vaginal products. Our group has previously reported that DPV is a substrate of cytochrome P450 enzymes 1A1 and 3A4 and that the activity of these enzymes was substantially inhibited by the combination of DPV and MIC [20]. Therefore, during concomitant use of DPV and MIC, the metabolism of DPV will be markedly reduced, leading to more parent DPV entering systemic circulation, as observed in IPM 028 [19].

In addition to drug metabolizing enzymes, drug transporters may also affect drug disposition, and the interactions of drug transporters with the combination of DPV and MIC have not been previously investigated. Transporters are known as determinants for drug disposition and factors for introducing drug-drug interactions [21]. Therefore, it is required by both the U.S. Food and Drug Administration (FDA) and the EMA to study potential interactions between transporters and investigational drugs. In guidance documents from both FDA and EMA, it is recommended to study interaction regarding P-glycoprotein (P-gp), breast cancer resistance protein (BCRP), organic cation transporters (OCTs), and organic anion transporters (OAT/OATPs), if applicable [22,23]. It is reported that expression levels of OCTs and OAT/OATPs in the human female reproductive tract (FRT) are weak or not detectable [24,25,26]. Therefore, these transporters are unlikely to have any impact on the observed DPV-MIC interaction. In contrast, substantial expression of P-gp and BCRP in the epithelium of human FRT has been confirmed [24,25,26,27]. In vitro evidence of MIC inhibiting P-gp activity has also been reported [28,29]. Other antifungal drugs in the azole family, including itraconazole, posaconazole, and isavuconazole, have been reported to inhibit the activity of P-gp or BCRP [30,31,32,33]. Additionally, there are other transporters highly expressed in the human FRT, including two multidrug resistance proteins (MRP1 and MRP4) and equilibrative nucleoside transporter 1 (ENT1), with confirmed interaction with antiretroviral drugs [24,26,34,35]. However, little is known about the interaction between DPV and transporters. Since DPV has poor solubility and high hydrophobicity (logP: 5.6), it is potentially a class 2 or 4 drug according to the biopharmaceutics classification system [36] and thus, the disposition of DPV can potentially be affected by drug transporters. If DPV were to be identified as a substrate of drug transporters that are highly expressed in human FRT, modulation on the activity of these transporters could affect DPV absorption in local human FRT tissue and subsequently affect systemic concentrations of DPV.

Tight junctions consist of dynamic protein structures and maintain the mucosal barrier against drug permeation. Tight junctions are also present in the epithelium of human FRT [37]. Disrupting tight junctions has been reported to elevate paracellular permeability and therefore enhance drug absorption [38,39]. Thus, it is also important to evaluate the impact of DPV and MIC on tight junction integrity as its disruption can alter drug exposure.

Tissue models maintain physiological structures that are more similar to in vivo conditions when compared to in vitro cell-based models and therefore can generate more clinically relevant results. Previous studies by our group reveal that human cervical tissue is useful for studying drug permeability and tissue drug retention. Using freshly excised human cervical tissue, we have evaluated the effects of commonly used excipients on the permeability of known paracellular and transcellular markers to investigate if excipients affect tight junction integrity and fluidize cell membranes [40,41]. In addition, human cervical tissue has previously been used to study tissue permeability and tissue levels of DPV formulated in vaginal films and rings, as well as to support the development of a physiologically based pharmacokinetic model for vaginal dosage forms [42,43]. Therefore, studying the impact of MIC on DPV permeability using human cervical tissue has the potential to provide clinically relevant data to better understand potential interactions.

The purpose of this study was to investigate the transport and permeation properties of DPV to better understand the clinically observed increased systemic exposure of DPV with concomitant use of MIC. The interactions between DPV and five drug transporters were evaluated using in vitro systems. The impact of DPV and MIC on tight junction was evaluated with cellular systems. Furthermore, the effect of MIC on DPV permeability was evaluated with human cervical tissues.

## 2. Materials and Methods

### 2.1. Materials

DPV (CAS number- 244767-67-7) and ^3^H-DPV (6.1 Ci mmol^−1^) were kindly provided by IPM (Silver Spring, MD, USA). Methanol (HPLC grade), sodium chloride, calcium hydroxide, glucose, 1N hydrochloric acid, 10% buffered formalin, xylene, ethanol, paraffin embedding wax, and 30% hydrogen peroxide were obtained from Fisher Scientific (Hampton, NH, USA). 50 mg mL^−1^ geneticin (G418 sulfate), fetal bovine serum, 100X penicillin-streptomycin-glutamine, Hank’s balanced salt solution (HBSS), dimethyl sulfoxide, digoxin, SuperScript IV First-Strand Synthesis system, 20% SDS, Pierce BCA protein assay kit, β-actin (15G5A11/E2) antibody, rabbit anti-mouse IgG secondary HRP antibody, human MRP1 and MRP4 vesicles, control vesicles for ABC transporters, MRPs-BCRP vesicular transport assay reagent set, and Cytoseal 60 were obtained from Thermo Fisher Scientific (Waltham, MA, USA). Dulbecco’s modified Eagle medium, minimum essential medium, 10x phosphate-buffered saline, Pierce lysis buffer, 0.25% trypsin and 0.1% ethylenediaminetetraacetic acid (EDTA) in HBSS, and Dulbecco’s phosphate-buffered saline were obtained from Corning Inc. (Corning, NY, USA). Primers for qPCR were obtained from Integrated DNA Technologies (Coralville, IA, USA). pCMV6 vectors were obtained from OriGene (Rockville, MD, USA). FuGENE 6 transfection system and ADP-Glo Max assay kit were obtained from Promega (Madison, WI). Calcium chloride dihydrate, potassium hydroxide, acetic acid glacial, lactic acid, and urea were obtained from Spectrum Chemical (New Brunswick, NJ). Verapamil, Ko143, dithiothreitol, nitrofurantoin, nuclease-free water, MIC, ethylene-bis(oxyethylenenitrilo)tetraacetic acid (EGTA), nitrobenzylthioinosine, sodium bicarbonate, glutathione, N-ethylmaleimide, prostaglandin E_2_, Tris-MOPS buffer, Krebs-Ringer buffer powder, estradiol 17 β-D-glucuronide, glycerin, and bovine serum albumin were obtained from Sigma-Aldrich (St. Louis, MO, USA). Ultima gold scintillation cocktail, ^3^H-estradiol 17 β-D-glucuronide (45.7 Ci mmol^−1^), ^3^H-digoxin (26.3 Ci mmol^−1^), ^3^H-thymidine (6.7 Ci mmol^−1^), ^14^C-mannitol (56.4 mCi mmol^−1^), and Solvable were obtained from PerkinElmer (Waltham, MA, USA). ^3^H-nitrofurantoin (20 Ci mmol^−1^) was obtained from American Radiolabeled Chemicals (St. Louis, MO, USA). MK571 was obtained from Selleck Chemicals LLC (Houston, TX, USA). 2x Ssofast Evagreen supermix, 10x Tris/Glycine/SDS buffer, tween 20, Clarity Max Western ECL substrate, and nonfat dry milk were obtained from Bio-Rad (Hercules, CA, USA). ENT1 (F-12) antibody was obtained from Santa Cruz Biotechnology, Inc. (Dallas, TX, USA). The MDCKII cell lines were provided by Dr. Philip E. Empey (University of Pittsburgh, Pittsburgh, PA, USA). HEK293 wild-type cells (HEK293 WT), HEC-1-A cells and McCoy’s 5A medium were obtained from ATCC (Manassas, VA, USA). Hematoxylin and Eosin staining kit was obtained from Vector Laboratories, Inc. (Burlingame, CA, USA).

### 2.2. Vesicular Uptake in Membrane Vesicles

To investigate if DPV was a substrate or inhibitor of MRP1 or MRP4, ATP-dependent and independent uptake of DPV (in the substrate study) or estradiol 17 β-D-glucuronide (a known MRP1/MRP4 substrate in the inhibitor study) was measured using 50 μg MRP1, MRP4, and control vesicles. MK571 was used to inhibit MRP1 and MRP4 activity in the substrate study and as a positive control in the inhibitor study [44]. The uptake assay was performed at a total volume of 50 μL with either 4 mM ATP or AMP, 2 mM glutathione, 10 μM DPV or estradiol 17 β-D-glucuronide as substrates, with and without 20 μM DPV or MK571 as inhibitors. The DPV concentration tested was determined based on DPV solubility limit in the assay buffer. To quantify the uptake amount of DPV or estradiol 17 β-D-glucuronide, 4 μCi mL^−1 3^H-DPV or ^3^H- estradiol 17 β-D-glucuronide was added to the assay mixture.

The vesicular uptake assays were performed under the above-mentioned conditions. The membrane vesicles were first incubated with respective inhibitor solution or blank assay buffer. The reaction was then initiated by adding the remaining components of the assay mixture to the vesicles and incubated at 37 °C for 3 min. To terminate the reaction, 200 μL ice-cold buffer was added to each well. The samples were then transferred to a glass fiber filter plate and washed 5 times by rapid filtration using a vacuum manifold (IBI Scientific, Dubuque, IA, USA). To quantify ^3^H-DPV and ^3^H-estradiol 17 β-D-glucuronide uptake, the washed vesicles were lysed with scintillation cocktail at room temperature for 20 min. The collected samples were analyzed on an LS6500 liquid scintillation counter (Beckman Coulter, Brea, CA, USA).

If ATP-dependent uptake was observed, the uptake amount by active transport was calculated using Equation (1):(1)$$\mathrm{Normalized}\;\mathrm{uptake}=\left(\mathrm{mean}\;\mathrm{uptake}\;\mathrm{with}\;\mathrm{ATP}\right)-\left(\mathrm{mean}\;\mathrm{uptake}\;\mathrm{with}\;\mathrm{AMP}\right)$$

### 2.3. ATPase Assay

ATPase activity of MRP1 or MRP4 in transporter overexpressing membrane vesicles (10 μg of protein per assay) was measured by the ADP-Glo Max assay according to manufacturer’s protocol. The assay quantified the amount of ADP produced by the membrane vesicles over 60 min at 37 °C in the ATPase assay buffer (50 mM Tris-MOPS, 50 mM potassium chloride, 5 mM sodium azide, 2 mM dithiothreitol, 2 mM glutathione, 0.1 mM EGTA-Tris, and 10 mM magnesium chloride at pH 7.0). DPV was tested at 20 μM and compared to the baseline ATPase activity. A known MRP1 substrate, N-ethylmaleimide-S-glutathione (prepared by mixing N-ethylmaleimide and glutathione at a 1:1 molar ratio), at 10 mM and a known MRP4 substrate, prostaglandin E_2_, at 20 μM were used as positive controls. The reaction was initiated with 5 mM ATP and quenched with the ADP-Glo detection buffer. The amount of ADP produced was quantified by recording luminescence on a SpectraMax M3 plate reader (Molecular Devices, San Jose, CA, USA).

### 2.4. Cell Culture

Wild-type and P-gp overexpressing MDCKII cells (MDCKII WT and MDCKII P-gp) were cultured in Dulbecco’s modified Eagle medium supplemented with 10% fetal bovine serum and 1x penicillin-streptomycin-glutamine. BCRP overexpressing and empty-vector containing MDCKII cells (MDCKII BCRP and MDCKII WT/EV) were cultured in Dulbecco’s modified Eagle medium supplemented with 10% fetal bovine serum, 800 μg mL^−1^ G418 sulfate, and 1x penicillin-streptomycin-glutamine. ENT1 overexpressing and empty-vector containing HEK293 cells (HEK293 ENT1 and HEK293 EV) were made by transfecting HEK293 WT cells with either pCMV6-ENT1 vector or pCMV6-Entry vector using the FuGENE 6 system following the manufacturer’s instructions. The transfected HEK293 cells were cultured in minimum essential medium supplemented with 10% fetal bovine serum, 500 μg mL^−1^ G418 sulfate, and 1x penicillin-streptomycin-glutamine. HEC-1-A cells were cultured in McCoy’s 5A medium supplemented with 10% fetal bovine serum and 1x penicillin-streptomycin-glutamine. All cells were cultured at 37 °C with 5% CO_2_. Cells were cultured to 80% confluence and then harvested by trypsinization; MDCKII cells were incubated with 0.25% trypsin and 0.1% EDTA in HBSS at 37 °C for 10 min, whereas HEK293 and HEC-1-A cells were incubated with 0.25% trypsin and 0.1% EDTA in HBSS at 37 °C for 5 min. Trypsinization was then terminated by adding equal volume of serum-containing culture media. The collected cell suspension was centrifuged at 125 RCF for 6 min. The cell pellets were then resuspended in serum-containing culture media and seeded to culture flasks or plates.

### 2.5. Bi-Directional Transport across MDCKII Cell Monolayers

MDCKII cells were seeded at a density of 10,000 cells per well onto 0.4 μm permeable transwell inserts in 12-well plates (Corning Inc., Corning, NY, USA). Cells were cultured and allowed to differentiate for 3 to 6 days. The cell growth media was changed every other day and the day before the transport assay. The quality of cell monolayers grown on the permeable inserts was evaluated by measuring transepithelial electrical resistance (TEER) using a Millicell-ERS apparatus (Millipore, Bedford, MA, USA). Only monolayers displaying TEER values above thresholds that indicated confluence (380 Ωcm^2^ for MDCKII P-gp cells and 175 Ωcm^2^ for other MDCKII cells) [45,46,47] were used in this study.

Bidirectional transport assays, including apical-to-basolateral (A to B) transport and basolateral-to-apical (B to A) transport, were performed with the qualified MDCKII monolayers. MDCKII P-gp and WT cells were used to perform the P-gp study, while MDCKII BCRP and WT/EV cells were used to perform the BCRP study. The buffer for bidirectional transport assays was HBSS at pH 7.4. To evaluate if DPV was a substrate of P-gp/BCRP, DPV was investigated at 0.5 μM with and without known transporter inhibitors (100 μM verapamil to inhibit P-gp and 1 μM Ko143 to inhibit BCRP). 0.5 μCi mL^−1 3^H-DPV was added for detection. To evaluate if DPV inhibited the activity of P-gp/BCRP, transport of known substrates (1 μM digoxin for P-gp study and 1 μM nitrofurantoin for BCRP study) was investigated with and without 5 μM DPV. The DPV concentration tested was determined based on DPV solubility limit in the assay buffer. 0.5 μCi mL^−1 3^H-digoxin or 0.5 μCi mL^−1 3^H-nitrofurantoin was used for detection. 0.25 μCi mL^−1 14^C-mannitol was used as a paracellular marker to confirm the tightness of the cell monolayers. All drugs were first dissolved in dimethyl sulfoxide and then diluted with HBSS. The final dimethyl sulfoxide concentration in each solution was less than 0.1%.

Before each experiment, the monolayers were washed with HBSS. Inhibitor solution or blank HBSS was then added to the donor side for treated groups or control groups, while the receptor side was incubated with blank HBSS. The volume of the apical chamber was 0.5 mL and that of the basolateral chamber was 1.5 mL. After a 30-min incubation at 37 °C, solution at the donor side was replaced with drug solution with or without respective inhibitors to initiate the transport assay. Cells were incubated with test articles for 90 min in a PR1MA mini-incubating orbital shaker (Midwest Scientific, Fenton, MO, USA) at 37 °C and 60 rpm (equivalent to 0.76 RCF). Samples from the donor side were collected at 0 and 90 min and the same volume of drug solution was replenished. Samples from the receptor side were collected every 15 min for up to 90 min and the same volume of blank HBSS was replenished after each sampling. The collected samples were quantified on an LS6500 liquid scintillation counter. Only data collected from cell monolayers with less than 1% of donor mannitol transferred per hour were used for subsequent calculation.

The apparent permeability coefficient (P_app_) was calculated using Equation (2):(2)$$P_{\mathrm{app}}=\frac{\mathrm{dQ}/\mathrm{dt}}{A\times C_{0}}\;$$
where dQ/dt, representing the initial linear flux, is the slope obtained from the linear regression plot between Q (cumulative amount transported to the receptor side) and t (time), A is the surface area of exposure, and C_0_ is the initial drug concentration at the donor side. Then the efflux ratio (ER) was calculated using Equation (3):(3)$$\mathrm{ER}=\frac{{\mathrm{mean}\;\mathrm{of}\;P}_{\mathrm{app}}\;B\;\mathrm{to}\;A}{{\mathrm{mean}\;\mathrm{of}\;P}_{\mathrm{app}}\;A\;\mathrm{to}\;B}$$
with the ER values, the net flux ratio was calculated using Equation (4):(4)$$\mathrm{Net}\;\mathrm{flux}\;\mathrm{ratio}=\frac{\mathrm{ER}\;\mathrm{of}\;\mathrm{overexpressing}\;\mathrm{cell}\;\mathrm{line}}{\mathrm{ER}\;\mathrm{of}\;\mathrm{control}\;\mathrm{cell}\;\mathrm{line}}$$

### 2.6. qPCR Analysis

Total RNA from HEK293 cells was extracted with the RNeasy Mini Kit (Qiagen, Hilden Germany). cDNA samples were made by reverse transcription with the SuperScript IV First-Strand Synthesis system on an MJ Mini thermal cycler (Bio-Rad, Hercules, CA, USA). The expression level of ENT1 was quantified via qPCR on a CFX96 real-time PCR detection system (Bio-Rad, Hercules, CA, USA) with our reported method [26,48]. The primer sequences are shown in Table 1. Each 10 μL qPCR reaction mixture contained 10 ng μL^−1^ cDNA template, 0.4 μM forward and reverse primers, and 1x Ssofast EvaGreen Supermix. The qPCR started with 30 s of initial denaturation at 95 °C, followed by 40 cycles of 5 s of denaturation at 95 °C and 30 s of annealing/extension at 60 °C. The relative expression level of ENT1 was calculated using the 2^−ΔΔCq^ method in the Bio-Rad CFX Manager software, with GAPDH as the reference gene.

### 2.7. Western Blot Analysis

Proteins from cell lysates of HEK293 ENT1 and HEK293 EV cells were quantified with the Pierce BCA protein assay kit on a SpectraMax M3 plate reader (Molecular Devices, San Jose, CA, USA) and the protein lysates were diluted to a final concentration of 3 mg mL^−1^. The diluted proteins were fractionated on a Mini-PROTEAN precast gel (Bio-Rad, Hercules, CA, USA) at 100 V for 30 min followed by 150 V for another 30 min. The fractionated proteins were then transferred to a nitrocellulose membrane at 100 V for 1 h. After 1-h blocking in phosphate-buffered saline with pH 7.4 with 3% nonfat dry milk, the membrane was incubated overnight under 4 °C with primary antibodies (1:100 for anti-ENT1 and 1:10,000 for anti-β-actin). The following day, the membrane was washed and incubated with the secondary antibody (1:10,000, rabbit anti-mouse IgG-HRP secondary antibody) for 1 h at room temperature. Proteins were then detected with the Clarity Max Western ECL substrate on a ChemiDoc Imager. The expression level of β-actin was used as the control.

### 2.8. Cellular Uptake in HEK293 Cells

HEK293 ENT1 and HEK293 EV cells were seeded at a density of 5 x 10^4^ cells per well in 24-well plates and cultured for 4 days. Before the uptake assay, cells were starved in serum-free minimum essential medium at 37 °C for 3 h to activate the ENT1 transporter [49]. The assays were performed in Krebs-Ringer buffer with pH 7.4 supplemented with 25 mM sodium bicarbonate and 1.2 mM calcium chloride dihydrate. To evaluate if DPV was a substrate of ENT1, DPV was investigated at 0.5 μM with and without the ENT1 inhibitor, nitrobenzylthioinosine, at 200 nM. 0.3 μCi mL^−1 3^H-DPV was added for detection. To evaluate if DPV inhibited the activity of ENT1, uptake of the known ENT1 substrate, thymidine, was evaluated at 5 nM (0.3 μCi mL^−1 3^H-thymidine was used to achieve equivalent concentration) with and without 5 μM DPV. 200 nM nitrobenzylthioinosine was also used as the positive control for ENT1 inhibition. All drugs were first dissolved in dimethyl sulfoxide and then diluted with Krebs-Ringer buffer. The final dimethyl sulfoxide concentration in each solution was less than 0.1%.

Starved cells were first incubated with respective inhibitor solution or blank Krebs-Ringer buffer for 30 min at 37 °C. The uptake assay was initiated by replacing the incubation solution with drug solution with or without respective inhibitors and incubated at 37 °C for 10 min. To terminate the assay, ice-cold Dulbecco’s phosphate-buffered saline was added to cells. Cells were then washed twice with ice-cold Dulbecco’s phosphate-buffered saline and lysed with 1% SDS at room temperature for 20 min. Protein content from lysates was quantified with the Pierce BCA protein assay kit on a SpectraMax M3 plate reader and cellular drug content was quantified on an LS6500 liquid scintillation counter. The drug uptake was normalized against protein content. Changes in drug uptake between groups were calculated using the mean uptake values of individual groups.

### 2.9. Evaluation of Tight Junction Disruption

HEC-1-A cells were seeded at a density of 9000 cells per cm^2^ onto 0.4 μm permeable snapwell inserts (Corning Inc., Corning, NY, USA). Cells were cultured and allowed to differentiate for 6 to 8 days. The quality of cell monolayers grown on the permeable inserts was evaluated by TEER using a Millicell-ERS apparatus. Only monolayers displaying TEER values above the threshold indicating confluency (130 Ωcm^2^) [50] were considered qualified to be used in this study.

To evaluate tight junction disruption, qualified HEC-1-A cell monolayers were mounted onto the Ussing Chamber apparatus (Physiologic Instruments, San Diego, CA, USA) to monitor TEER values. After equilibrating cells with HBSS at 37 °C for 20 min, buffer in basolateral chambers was replaced with HBSS without calcium or magnesium (HBSS Ca^−^&Mg^−^), while buffer in the apical chambers was replaced with respective treatment solution in HBSS Ca^−^&Mg^-^. Treatments included 5 μM DPV, 5 μM MIC, and 5 μM DPV with 5 μM MIC. 2 mM EGTA, a known tight junction disruptor that decreases TEER, was used as the positive control and blank HBSS Ca^-^&Mg^-^ was used as negative control. Cells were incubated at 37 °C for 80 min and the obtained TEER values were normalized against initial measurements at time 0.

### 2.10. Evaluation of DPV Tissue Permeability

Freshly excised human ectocervical tissue samples from 6 pre-menopausal women were obtained through the Pitt Biospecimen Core at the University of Pittsburgh under an IRB-approved protocol (IRB# PRO09110431). Tissue samples were immersed in ice-cold Dulbecco’s modified Eagle medium during transfer from the surgery site to the laboratory. Tissue samples were then processed as previously reported [43], where excess stromal tissue was removed using a scalpel, and the epithelium was isolated using a Stadie-Riggs tissue slicer. The isolated epithelium sections were then mounted onto the in-line cell apparatus (PermeGear, Cranford, NJ, USA), a flow-through type Franz cell, with the epithelial side facing the apical compartment. A small piece of tissue sample was fixed in 10% formalin as the pre-treatment control for histology evaluation.

Vaginal fluid simulant was used to prepare drug solutions. Vaginal fluid simulant was prepared using a reported formulation (3.51 g L^−1^ sodium chloride, 1.40 g L^−1^ potassium hydroxide, 0.22 g L^−1^ calcium hydroxide, 0.018 g L^−1^ bovine serum albumin, 2.00 g L^−1^ lactic acid, 1.00 g L^−1^ acetic acid, 0.16 g L^−1^ glycerin, 0.4 g L^−1^ urea, and 5.0 g L^−1^ glucose) and adjusted to pH 4.2 with 1N hydrochloric acid [51]. To evaluate the impact of MIC on DPV tissue permeability, 450 μL of a mixture containing 182 µM (60 μg g^−1^) DPV and 63 mM (30 mg g^−1^) MIC was added to the apical compartment and compared to 182 µM (60 μg g^−1^) DPV alone. DPV and MIC were tested at the highest concentration levels in vaginal fluid reported from the IPM study (IPM 028) [19]. 2.5 μCi g^−1 3^H-DPV was added for quantification and 3.3 μCi g^−1 14^C-mannitol was used as a paracellular marker. Dulbecco’s modified Eagle medium was used in the basolateral compartment at a flow rate of 50 μL min^−1^. 1.5 mL samples (flow-through Dulbecco’s modified Eagle medium) were collected from the basolateral compartment at 30, 60, 90, 120, 150, 180, 210, 240, 270, 300, 330, and 360 min. 100 μL of drug suspension in the apical compartment was centrifuged at 2000× *g* for 2 min and 50 μL of supernatant was used to determine donor drug concentration. At the end of the permeability assay, tissue was washed in 1 mL ice-cold Dulbecco’s modified Eagle medium 6 times and then cut into two halves and weighed separately. One half of the tissue sample was fixed in 10% formalin for subsequent histological evaluation, while the other half was used for drug quantification. To quantify drug tissue levels, each tissue sample was digested overnight in 2 mL Solvable held at 50 °C. 200 μL of 30% hydrogen peroxide was then added and samples were incubated for an additional 30 min at 50 °C. Samples were then cooled to room temperature and 1.5 mL of each sample was used for quantification. To quantify ^3^H-DPV and ^14^C-mannitol, each vial of collected sample was mixed with 2 mL scintillation cocktail and analyzed on an LS6500 liquid scintillation counter. P_app_ was calculated using Equation (2) as described above.

### 2.11. Histology

Gross morphology of tissue samples was evaluated using hematoxylin and eosin (H&E) staining. After fixation in formalin, tissue samples were dehydrated in alcohol, cleared in xylene, and embedded in paraffin wax [52]. Five-micron sections of embedded tissue were used to perform H&E staining following manufacturer’s procedures (hematoxylin for 5 min, bluing reagent for 15 s, and eosin for 2 min) and mounted onto microscope slides using Cytoseal 60. Images were acquired using the AxioCam software, version 4.9.1 (Carl Zeiss, San Diego, CA, USA) paired with the microscope, Axioskop 40 (Carl Zeiss, San Diego, CA, USA).

### 2.12. Statistical Analysis

Statistical analysis was performed with unpaired Student’s *t*-test using GraphPad Prism, version 9.0 (GraphPad Software, La Jolla, CA, USA). A *p*-value less than 0.05 was considered statistically significant.

### 2.13. Study Quality Management

While these studies were not conducted in compliance with Good Laboratory Practice (GLP) 21 CFR Part 58, all experiments were conducted in compliance with approved protocols, quality management plans, and standard operating procedures.

## 3. Results

### 3.1. Evaluation of the Interaction between DPV and MRP1/MRP4

MRP1 and MRP4 are both efflux transporters in the Multidrug Resistance Associated Protein family. They are highly expressed in human FRT, and evidence of MRP4 activity limiting substrate absorption in FRT tissue in vivo has been reported [27,35]. Here, we used transporter-overexpressing membrane vesicles to study the potential interactions between DPV and MRP1/MRP4. Vesicular uptake of DPV and a known substrate of MRP1 and MRP4, estradiol 17 β-D-glucuronide, is shown in Figure 1. The uptake of DPV with ATP in either MRP1 or MRP4 vesicles was similar to that with AMP, indicating DPV uptake was ATP-independent. DPV uptake in MRP vesicles was not significantly different from the uptake in control vesicles and not affected by the known inhibitor of MRP1 and MRP4, MK571 (Figure 1A,B). The uptake of estradiol 17 β-D-glucuronide with ATP was significantly different from that with AMP, indicating estradiol 17 β-D-glucuronide uptake was ATP-dependent. The values of normalized estradiol 17 β-D-glucuronide uptake were 36.83 fmol by MRP1 vesicles, 35.83 fmol by MRP1 vesicles co-incubated with DPV, and 13.02 fmol by MRP1 vesicles co-incubated with MK571. The active uptake of estradiol 17 β-D-glucuronide by MRP1 vesicles was not significantly affected by 20 μM DPV but reduced to 35% by 20 μM MK571 (Figure 1C). The values of normalized estradiol 17 β-D-glucuronide uptake were 15.65 fmol by MRP4 vesicles, 14.08 fmol by MRP4 vesicles co-incubated with DPV, and 4.60 fmol by MRP4 vesicles co-incubated with MK571. The active uptake of estradiol 17 β-D-glucuronide by MRP4 vesicles was not significantly affected by 20 μM DPV but reduced to 29% by 20 μM MK571 (Figure 1D). ATPase activity in MRP1 vesicles was stimulated to 240% ± 7.4% of baseline by N-ethylmaleimide-S-glutathione, and that in MRP4 vesicles was stimulated to 187% ± 3.3% of baseline by prostaglandin E_2_. In contrast, DPV had no significant stimulation on the ATPase activity in either MRP1 (*p* = 0.75) or MRP4 (*p* = 0.99) vesicle (Figure 1E,F). Please note the pmol or fmol levels of test articles shown in the y-axis of all figures only reflect the concentrations of the radiolabeled portion of a test article.

### 3.2. Evaluation of the Interaction between DPV and P-gp/BCRP

P-gp and BCRP are the two most reported efflux transporters responsible for limiting substrate absorption and therefore, the investigation of potential interactions between drug candidates and P-gp/BCRP is recommended by both EMA and FDA [22,23]. High expression levels of P-gp and BCRP in human FRT has also been confirmed [24,25,26]. Here, we studied the potential interactions between DPV and P-gp/BCRP using the MDCKII cell line, a widely used epithelial cell model. In all Transwell transport assays, mannitol was used as a paracellular marker to confirm the integrity of cell monolayers. The linear transport of DPV across MDCKII P-gp and WT cell monolayers is shown in Figure 2. The flux of DPV across MDCKII WT cell monolayers showed few directional differences and was not affected by the transporter inhibitor, verapamil (Figure 2A). This suggested that endogenous efflux activity in MDCKII WT cells did not affect DPV transport, providing a low baseline efflux. In MDCKII P-gp cells, the B-to-A flux of DPV was slightly greater than the A-to-B flux, but not statistically significant (*p* = 0.065). A weak inhibitory effect from 100 μM verapamil was also observed as the bi-directional flux of DPV became closer, indicating a smaller directional difference of DPV permeability and the compromised efflux activity of P-gp (Figure 2B). The DPV P_app_ values for WT cells showed no significant difference between control and verapamil treated cells (Figure 2C), whereas verapamil significantly reduced the B-to-A DPV P_app_ for P-gp cells (Figure 2D). The calculated net flux ratio was 1.31. The ER in MDCKII P-gp cells was reduced to 65% by verapamil. No significant difference was observed in the A to B P_app_ and B to A P_app_ of mannitol with and without the presence of verapamil (Appendix A).

The linear transport of DPV across MDCKII BCRP and WT/EV cell monolayers is shown in Figure 3. The flux of DPV showed few directional differences and was not affected by the inhibitor, Ko143 (Figure 3A,B). In addition, the DPV P_app_ values had no directional difference and were not affected by Ko143 (Figure 3C,D). The calculated net flux ratio was 1.33. The ER in MDCKII BCRP cells was reduced to 85% by Ko143. No significant difference was observed in the A to B P_app_ and B to A P_app_ of mannitol with and without the presence of Ko143 (Appendix A).

The impact of DPV on the linear transport of known P-gp and BCRP substrates is shown in Figure 4. Across MDCKII P-gp or BCRP cell monolayers, the B-to-A flux of digoxin or nitrofurantoin was greater than the A-to-B flux, indicating the active efflux of the known substrates by P-gp or BCRP (Figure 4A,B). The active efflux of digoxin and nitrofurantoin was also supported by the observation that the A-to-B P_app_ values were significantly (*p* < 0.05) lower than the B-to-A P_app_ values. These were because both P-gp and BCRP were located at the apical side of MDCKII cells, resulting in active B-to-A transport of their substrates. When verapamil was added to inhibit P-gp, the B-to-A flux and P_app_ of digoxin in MDCKII P-gp cells decreased while its A-to-B flux and P_app_ increased (Appendix A). Similar effects in the flux and P_app_ of nitrofurantoin in MDCKII BCRP cells were observed when Ko143 was added to inhibit BCRP (Appendix A). Digoxin ER in MDCKII P-gp cells was reduced to 11% by verapamil, while nitrofurantoin ER in MDCKII BCRP cells was reduced to 34% by Ko143. These results support the use of our MDCKII cells to evaluate potential inhibitory effects on P-gp and BCRP. 5 μM DPV had no significant impact on P_app_ values of these known substrates from either direction, indicating that DPV at 5 μM did not inhibit the activity of P-gp and BCRP (Figure 4C,D). No significant difference was observed in the A to B P_app_ and B to A P_app_ of mannitol with and without the presence of verapamil or Ko143 (Appendix A).

### 3.3. Evaluation of the Interaction between DPV and ENT1

ENT1 is an uptake transporter that is highly expressed in human FRT [26]. Here, we first constructed HEK293 cells overexpressing ENT1 and empty-vector control cells. We then used these cells to study the potential interaction between DPV and ENT1. The overexpression of ENT1 in the transfected HEK293 cells is shown in Figure 5. At the mRNA level, the ENT1 expression level in HEK293 ENT1 cells was significantly higher than that in WT cells, while the ENT1 expression level in HEK293 EV cells was similar to that in WT cells (Figure 5A). At the protein level, the ENT1 expression level relative to the β-actin level was much higher in HEK293 ENT1 cells than that in HEK293 EV cells (Figure 5B). These results confirmed the successful construction of an ENT1 overexpressing cell line to study the interaction between DPV and ENT1. Cellular uptake of DPV in HEK293 ENT1 cells and that in HEK293 EV cells were similar. Neither cell type was significantly affected by the inhibitor, 200 nM nitrobenzylthioinosine (Figure 5C). In HEK293 ENT1 cells, cellular uptake of a known ENT1 substrate, thymidine, was reduced to 30% by 200 nM nitrobenzylthioinosine, while 5 μM DPV had no significant impact on thymidine uptake (Figure 5D).

### 3.4. Impact of DPV and MIC on Cellular Tight Junction

Tight junctions form a barrier for drug penetration in the human FRT epithelium. They are responsible for limiting passive drug diffusion [38,39]. Tight junction integrity can be monitored by TEER [50]. Here, we used an epithelial cell line from human FRT that stably formed tight junctions (HEC-1-A) to evaluate whether DPV and MIC affect tight junctions. After 40 min of treatment with the positive control EGTA, TEER values dropped significantly compared to the negative control, indicating tight junction disruption. By the end of the 80-min exposure, the relative TEER percentages measured from the EGTA groups were only 60.9 ± 9.6 of the baseline value. The mean relative TEER percentages at 80 min were 94.7 ± 7.8 with DPV alone, 81.6 ± 6.5 with MIC alone, and 77 ± 7.8 with DPV/MIC combination and they were not significantly different from those in their respective negative control groups (less than 1.5% difference). The gradual decrease of TEER values was also observed in the negative control group exposed to blank HBSS Ca^−^&Mg^−^, which was mainly due to the absence of extracellular calcium. Low extracellular calcium affected membrane association of occluding and zonula occludens proteins, resulting in decreased tightness of tight junctions that was observed as a decrease in TEER values [53].

### 3.5. Impact of MIC on DPV Tissue Permeability

The human vagina and ectocervix are covered by a stratified squamous epithelium that consists of multiple layers of epithelial cells, and such a structure is difficult to replicate in cellular models [54]. Excised human cervical tissue can be used to generate more clinically relevant results than cellular models because the in vivo structure is largely maintained. Here, we evaluated whether MIC affects DPV permeability in excised human cervical tissue at the highest DPV drug level (182 µM in suspension) detected in vaginal fluid in IPM 028 [19]. Permeability and tissue levels of DPV are shown in Figure 6. MIC had no significant impact on the amount of cumulative DPV permeated across human cervical tissue (*p* > 0.05 for all time points) (Figure 6A). Across human cervical tissue, DPV had a P_app_ value of 0.88 ± 0.53 × 10^−6^ cm s^−1^, which was not significantly (*p* = 0.28) affected by the addition of MIC (Figure 6B). However, MIC significantly (*p* < 0.05) reduced tissue levels of DPV by 59% (Figure 6C). In all replicates, less than 1% donor mannitol was permeated per hour, which indicated maintenance of tissue integrity during experiments [54,55]. Pre- and post-exposure histological evaluation indicated that all tissue samples retained intact epithelium after the permeability assay with similar morphology to the pre-treatment samples, where the stratified squamous epithelium remained tightly attached to the stroma and nuclei of epithelial cells could be clearly identified (Figure 6D–F). These results indicated that DPV and MIC at the tested concentration levels did not damage the human cervical tissue epithelial barrier or alter tissue morphology within the conditions of the permeability study.

## 4. Discussion

Transporters have been reported to greatly affect the tissue penetration of antiretroviral drugs [34,35,56]. For example, tenofovir, a substrate of the efflux transporter MRP4, showed low tissue penetration efficiency in the FRT and co-administration with the MRP4 inhibitor, MK571, significantly increased tissue penetration and systemic exposure of tenofovir [35,56]. In contrast, tissue penetration of abacavir was significantly facilitated by the uptake transporter ENT1 [34]. Our results suggest DPV is not a good MRP1, MRP4, BCRP, or ENT1 substrate and that these transporters have little impact on tissue penetration of DPV. The lack of substrate-type interactions with BCRP and MRP1 was also reported with other NNRTIs, including etravirine, efavirenz, and nevirapine [57]. Although asymmetric transport of DPV was observed across P-gp cell monolayers, DPV did not meet the P-gp substrate criteria from the FDA’s guidance document (net flux ratio > 2 and ER reduces to < 50% with a P-gp inhibitor) [58]. This lack of efflux effect was consistent with a previous study, where DPV was identified as a P-gp substrate using in silico methods but showed no significant interaction with P-gp in cellular studies [59]. Since the interaction between DPV and P-gp was weak in overexpressing cells, P-gp is unlikely to have clinically significant impact on DPV distribution in human FRT. Our tissue permeability results showed that MIC had no significant impact on DPV permeability across human cervical tissue, despite the known P-gp inhibitory effects of MIC [26,28]. This suggests that the clinically observed DPV-MIC interaction is not due to MIC effects on the activity of the evaluated transporters. In a clinical study using a vaginal ring containing both DPV and maraviroc, DPV showed better tissue penetration compared to maraviroc [60]. This might be due to the lack of interaction between DPV and efflux transporters in FRT, while maraviroc is a known substrate of P-gp [61]. Additionally, disease-associated modulation of transporter activity has been seen in vivo, and alterations in transporter expression levels are known to occur due to fluctuating estrogen levels during natural cycles [35,62]. However, since transporters highly expressed in human FRT showed little impact on DPV distribution, conditions affecting transporter activity or expression are unlikely to have any effect on vaginally administered DPV.

Combinations of multiple antiretroviral drugs have been widely used in the treatment of HIV/AIDS and drug-drug interactions due to transporters have been well reported [26,63]. Alterations in saquinavir exposure were noted in combination with atazanavir or lopinavir due to the inhibitory effect on efflux transporters [63]. Our results showed that DPV had no inhibitory effect on efflux transporters highly expressed in human FRT, which could make DPV acceptable for use with other compounds that are transporter substrates. In fact, a DPV-darunavir vaginal gel and a DPV-maraviroc vaginal ring have already undergone clinical investigation, where both darunavir and maraviroc are substrates of P-gp [26]. Since the reported DPV tissue concentration from a DPV ring clinical study (IPM 026) was 1.8 μM, which was below the levels tested in our work, the DPV ring is expected to have negligible impact on the active transport of darunavir or maraviroc [60]. In addition, concomitant use of the DPV ring and other products containing substrates of P-gp, BCRP, MRP1, MRP4, and ENT1 is unlikely to introduce additional interactions due to modulation on transporter activity. However, it is also reported that DPV at 10 μM upregulated MRP1 expression level while downregulating that of MRP4 [24]. In addition, induction of P-gp was observed with another NNRTI, nevirapine [64]. Although transporter induction was not evaluated in this study, the reported DPV tissue concentration is lower than the concentration required for altering transporter gene expression.

In addition to transporters, the multilayer epithelial barrier in human FRT is a physical barrier for drug permeability by limiting passive drug diffusion. An intact epithelial barrier is crucial for maintaining the health of human FRT organs. Optimal HIV prevention products should be safe to users and not impair the epithelial barrier of the local mucosa. The tight junctions are a major component of the human FRT epithelial barrier and protect the FRT against viral infection [65]. Our results showed that at 5 μM, DPV, MIC, or their combination had no disruptive effect on tight junctions. In addition, our histology results showed that DPV alone and the combination of DPV and MIC at clinically relevant concentrations did not alter the gross morphology of the human cervical epithelium after 6-h exposure. This is consistent with the excellent safety profile of DPV vaginal ring observed in clinical studies [4,10]. Therefore, concomitant use of the DPV ring and MIC suppository is unlikely to increase the chance of HIV infection, and the observed DPV-MIC interaction is not caused by tight junction disruption or alteration in local tissue epithelium integrity [18,65].

Our laboratory has previously evaluated the interactions between DPV and human cytochrome P450 (CYP) and uridine 5′-diphospho-glucuronosyltransferase (UGT) enzymes in human FRT, and evaluated the influence of MIC on these interactions [20]. CYP enzymes are expressed in the human FRT and MIC is known to have inhibitory effects on CYP enzymes [26,66,67]; increased exposure of CYP substrates due to CYP inhibition from MIC has been reported [68]. In addition, MIC has been shown to inhibit substrate glucuronidation by UGT enzymes, which are also expressed in the human FRT [26,69]. We found that DPV was a substrate of CYP1A1 and CYP3A4, but was not a substrate of UGT enzymes. We also showed that CYP enzyme activity is substantially inhibited by the combination of DPV and MIC [20]. The observed DPV-MIC interaction was therefore likely attributable to reduced DPV metabolism due to inhibition of CYP1A1 and CYP3A4 by MIC. In the tissue permeability study reported here, liquid scintillation counting was used and thus, the quantified drug-related material represented the total amounts of DPV-related material present in the tissue, including DPV and the metabolites. In tissue exposed to DPV alone, it was likely that the measured concentrations consisted of parent DPV and a substantial number of metabolites. The depletion of parent DPV through metabolism could create a concentration gradient allowing for the influx of further DPV, thus increasing the concentration of drug-related material. In tissue exposed to DPV and MIC, CYP enzyme activity in tissue was likely inhibited, in which case the measured concentration would comprise mainly parent DPV, and the potential for further influx of DPV would be limited, resulting in lower overall concentrations of DPV-related material. It is therefore possible that the apparent reduction in the tissue concentration of DPV observed in this study was due to the enzyme inhibition observed previously, since the lack of interaction between DPV and transporters highly expressed in human FRT and the absence of any tight junction disruption by DPV/MIC suggest that these factors were not implicated in the effect on tissue concentrations. These findings also provide crucial insights for the optimization of physiological based pharmacokinetic models to predict the pharmacokinetics and distribution of vaginally administered DPV. This is currently being evaluated through a separate study by our group. Since drug metabolizing enzymes, not drug transporters, are key factors affecting the pharmacokinetics of vaginally administered DPV, investigating the disposition of parent DPV as well as its metabolites in human FRT can improve existing physiologically based pharmacokinetic models for vaginal drug delivery when applied to DPV [42].

## 5. Conclusions

In summary, the current work provides insight into the role of transporters in the disposition of DPV in FRT, possible effects of DPV and MIC on tight junctions, and possible effects of MIC on DPV tissue permeability and tissue levels. Using vesicular and cellular systems, our study has shown that DPV is not a substrate of P-gp, BCRP, MRP1, MRP4, and ENT1 transporters. DPV does not have inhibitory effect at 5 μM on P-gp, BCRP and ENT1 or at 20 μM on MRP1 and MRP4. DPV or MIC at 5 μM and their combination did not result in tight junction disruption. At clinically relevant concentrations, MIC was not shown to affect DPV permeability across human cervical tissue but a significant reduction in tissue DPV levels was observed in this study. We further revealed that interactions between DPV and MIC are not due to modulation of transporters expressed in the FRT. This is the first report that has thoroughly evaluated the interactions between DPV and transporters expressed in the human FRT.

## Figures and Tables

**Figure 1 pharmaceutics-14-01948-f001:**
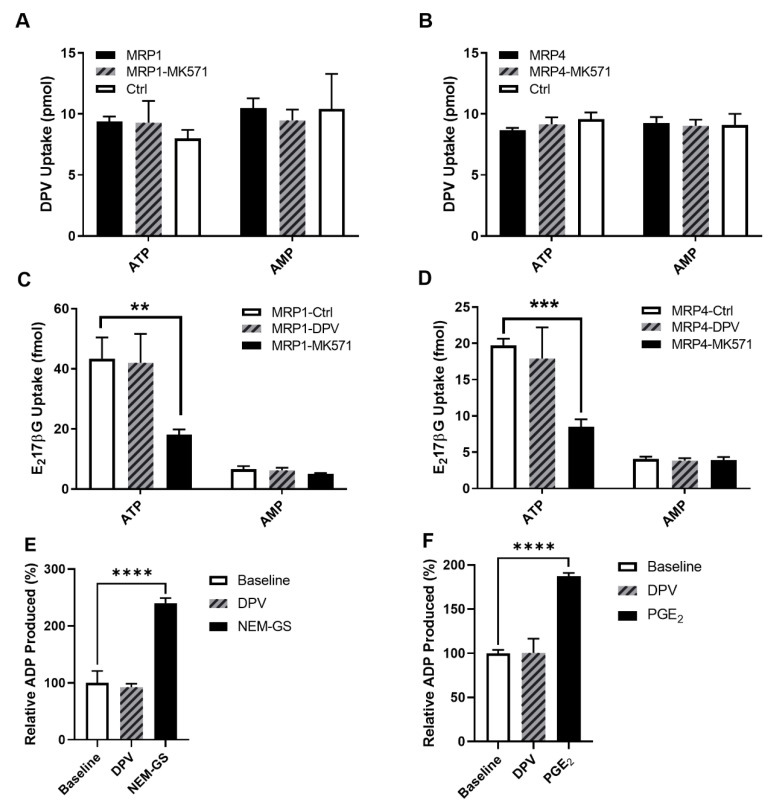
Vesicular uptake of DPV in MRP1 (**A**) and MRP4 (**B**) vesicles without and with MK571 and in control vesicles (ctrl); Vesicular uptake of estradiol 17 β-D-glucuronide (E_2_17βG) in MRP1 (**C**) and MRP4 (**D**) vesicles alone (ctrl), with DPV, and with MK571; ATPase activity in MRP1 (**E**) and MRP4 (**F**) vesicles with N-ethylmaleimide-S-glutathione (NEM-GS) and prostaglandin E_2_ (PGE_2_) being the positive control substrates. Data represented as mean±SD of three biological replicates. ** *p* < 0.01. *** *p* < 0.001. **** *p* < 0.0001 (Student’s *t*-test).

**Figure 2 pharmaceutics-14-01948-f002:**
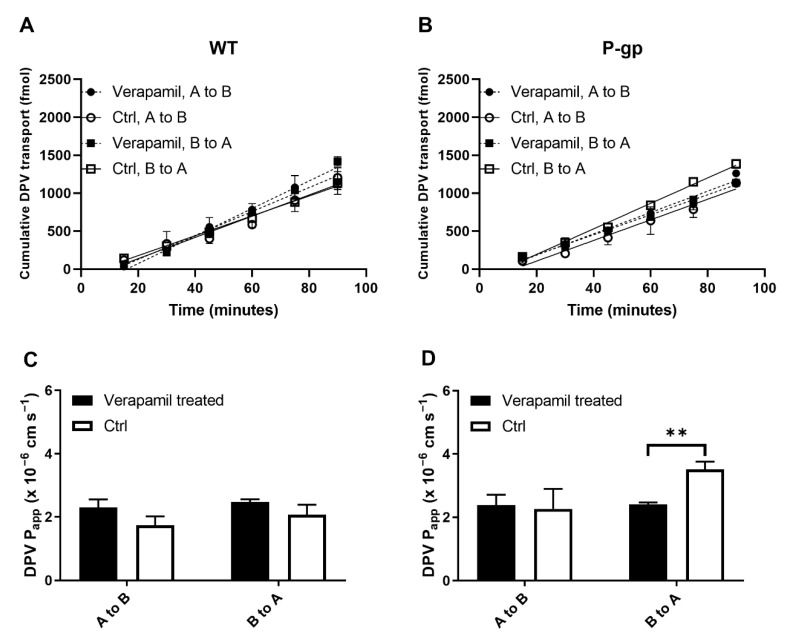
DPV bi-directional transport across MDCKII WT (**A**) and P-gp (**B**) cell monolayers with and without the transporter inhibitor, verapamil; P_app_ values obtained from the transport assays (**C**,**D**). Data represented as mean±SD of three biological replicates. ** *p* < 0.01 (Student’s *t*-test).

**Figure 3 pharmaceutics-14-01948-f003:**
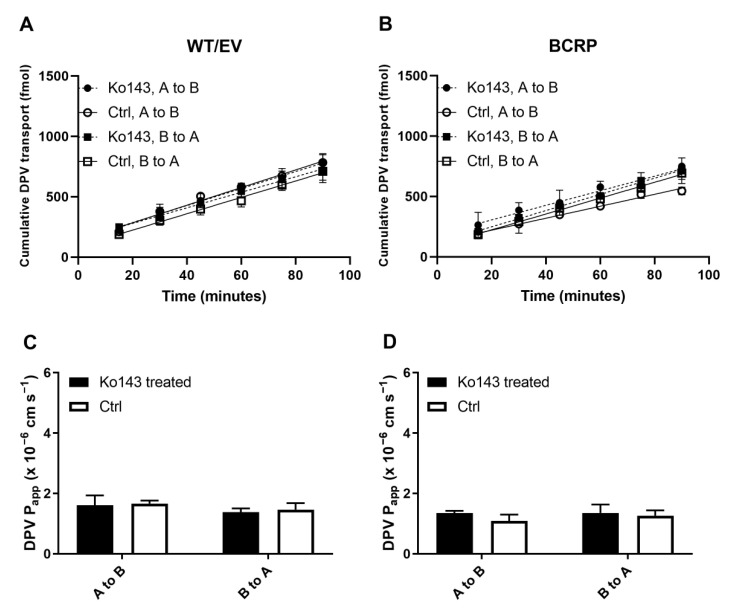
DPV bi-directional transport across MDCKII WT/EV (**A**) and BCRP (**B**) cell monolayers with and without transporter inhibitors (Ko143 and Ctrl); P_app_ values obtained from the transport assays (**C**,**D**). Data represented as mean±SD of three biological replicates.

**Figure 4 pharmaceutics-14-01948-f004:**
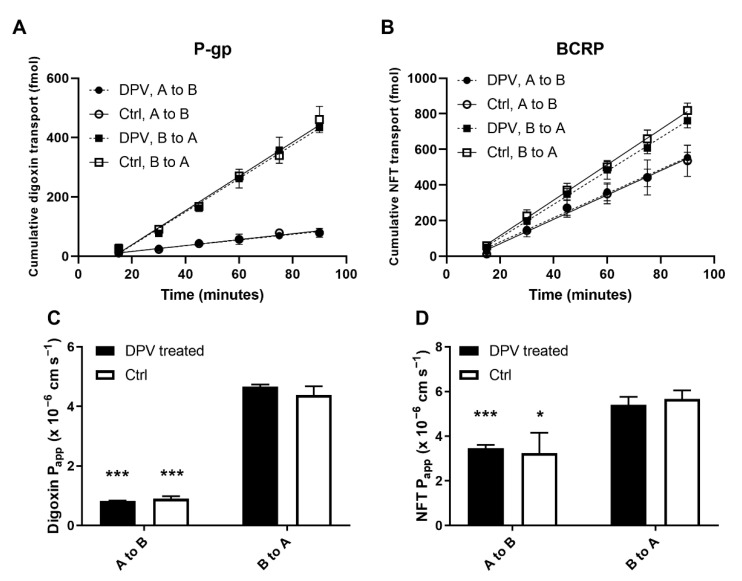
Bi-directional transport of digoxin across MDCKII P-gp (**A**) and nitrofurantoin (NFT) across MDCKII BCRP (**B**) cell monolayers with and without DPV (DPV and Ctrl); P_app_ values obtained from the transport assays (**C**,**D**). Data represented as mean±SD of three biological replicates. * *p* < 0.05 in comparison to B to A. *** *p* < 0.001 in comparison to B to A (Student’s *t*-test).

**Figure 5 pharmaceutics-14-01948-f005:**
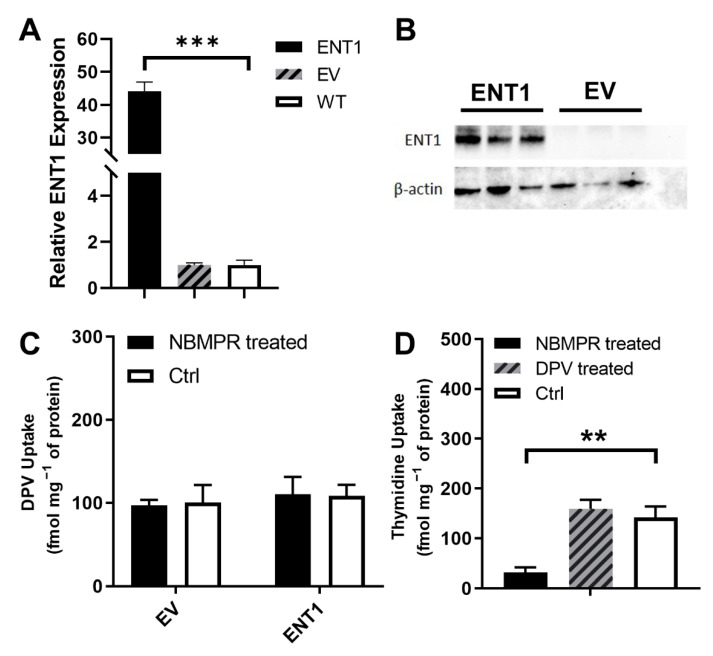
(**A**) Relative expression of ENT1 from HEK293 cells at mRNA level, *** *p* < 0.001 compared to WT; (**B**) Expression of ENT1 at protein level; (**C**) Normalized cellular uptake of DPV in HEK293 ENT1 and EV cells with and without nitrobenzylthioinosine (NBMPR and Ctrl) (*n* = 3); (**D**) Normalized cellular uptake of thymidine in HEK293 ENT1 cells treated with nitrobenzylthioinosine, DPV, or blank buffer (NBMPR, DPV, or Ctrl). Data represented as mean ± SD of three biological replicates. ** *p* < 0.01 in comparison to no drug treatment (Student’s *t*-test).

**Figure 6 pharmaceutics-14-01948-f006:**
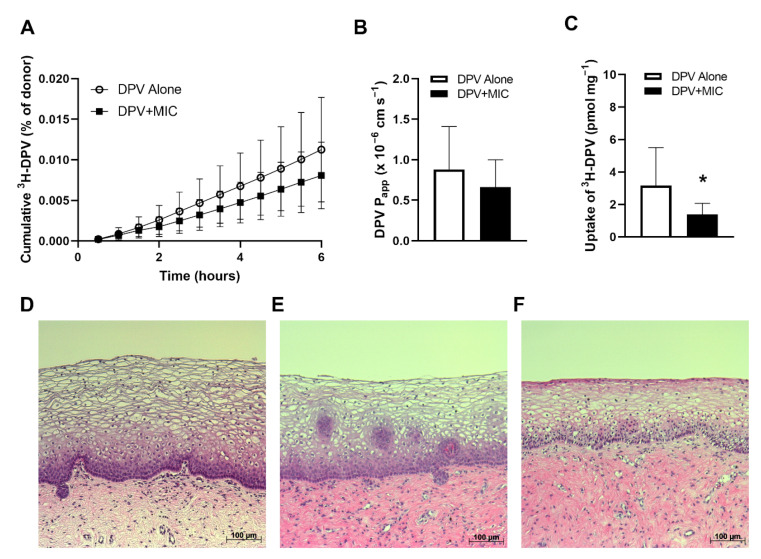
Linear flux of DPV across human cervical tissue (**A**), DPV P_app_ values obtained from the tissue permeability assay (**B**), and DPV levels in human cervical tissue (**C**), * *p* < 0.05 compared to DPV alone (Student’s *t*-test); Data represented as mean ± SD with *n* = 10–12 replicates from 6 tissue donors; Representative histology results of pre-treated tissue (**D**), tissue post-exposure to DPV alone (**E**), and tissue post-exposure to DPV and MIC (**F**).

**Table 1 pharmaceutics-14-01948-t001:** Primer sequences for qPCR analysis.

Common Name(Official Symbol)	GenBank Accession No.	Sequence of Primers (5′→3′)
GAPDH (GAPDH)	NM_001357943	Forward: GGA GCG AGA TCC CTC CAA AAT
		Reverse: GGC TGT TGT CAT ACT TCT CAT GG
ENT1 (SLC29A1)	NM_001372327	Forward: TGA GCG GAA CTC TCT CAG TG
		Reverse: GA GGT AGG TGA ATA ACA GCA GG

## Data Availability

Data supporting the reported results will be available with the corresponding author (Lisa C. Rohan).

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
