# Peer review of "Transport and Permeation Properties of Dapivirine: Understanding Potential Drug-Drug Interactions"

_pharmaceutics, 2022, doi:10.3390/pharmaceutics14091948_

Round 1
Reviewer 1 Report
Authors have studied DPV-MIC interactions due to DPV transport and permeation, through the interactions between DPV and MRP1, MRP4, P-gp, BCRP, and ENT1 transporters. Results obtained by the authors suggest that these five transporters did not affect to DPV-MIC interactions, DPV and MIC had no effect on tight junctions and MIC did not interfere with DPV permeability across human cervical tissue. These results are of great interest for the combined use of DPV and MIC, especially in the HIV field.
I recommend it for publication in this journal. However, some minor specifications and changes should be done:
1. Section “2.4. Cell culture”. Please, specify trypsinization conditions for MDCKII WT, MDCKII P-gp, MDCKII BCRP, MDCKII 195 WT/EV, transfected HEK293 and HEC-1-A cell lines.
2. Section “2.5. Bi-directional transport across MDCKII cell monolayers”. Please, change “60 rpm” for the appropriate Relative Centrifugal Force (RCF) or G-force.
3. Following IUPAC rules, change “ml” for “mL”, “μl” for “μL”, “g/L” for “g L-1”, “mg/ml” for “mg mL-1”, “ng/μL” for “ng μL-1”, “μg/ml” for “μg mL-1”, “μCi/ml” for “μCi mL-1”, “μl/min” for “μL min-1”, “cm/s” for “cm s-1”, “μg/g” for “μg g-1”, etc. , throughout the manuscript.
4. Change “hours” for “h”, “minutes” for “min” and “seconds” for “s” throughout the manuscript.
Author Response
Dear Reviewer,
Thank you for your comments and suggestions. Please find our point-to-point responses to each of your comments in the attached file.
Sincerely,
Ruohui Zheng

Reviewer 2 Report
Ruohui Zheng et al., did excellent investigations that has clinical potential for understanding disposition and PK-drug-drug interactions. Here are my comments to improve the quality and understanding of the works-
Major comments:
1. As DPV has been reported by the same authors and others, that the CYP450 and UGT enzymes plays a role on clearance; and MIC is a potent inhibitor of CYPs. Therefore, reviewer suggest including few of these examples in the introduction to understand its basic ADME/DMPK pathways and hypotheses for DDI investigations and as they described in the discussion to interpret the findings form this study.
2. Author needs to calculate absolute amount of radiolabel substrates (positive control or test article) from the assay (Results 3.1 and so on….), it is risky to conclude from CPM count. Specific activity of the radiolabel compound and normalized with protein is the key factor that may affect to individual well count or variability. Reviewer suggests calculating fmol or pmol for the data points for the uptake (y-axis in figure 1, 2, 3, 4).
3. Reviewer suggesting including few physicochemical properties of DPV in the introduction to understand why there was a possibility to investigate transporters-interactions with DPV. Such as whether it has limited permeability, solubility, clearance that doesn’t explained by enzyme’s liabilities etc.
4. Author needs to show or explain why they used only 200nM DPV in the experiments. Usually, in vitro experiment should contain both rad and cold substrate to provide sufficient concentrations for the receptor-environment. I am worried that the active and passive permeability (uptake or efflux) is not affected by this very low concentration.
Minor comments:
1. What do you mean by n=3 replicates in the figure legends? Is it mean from 3 independent experiments or triplicate in a same day assay?
2. Author need to add little more about MIC and DPV DDIs with other clinical drug or endogenous probe in the introduction and discussions.
Author Response

(The authors gave the same response as above.)

Reviewer 3 Report
The submitted manuscript evaluated the role of the several transporters in the disposition of dapivirine in female reproductive tract, influence of dapivirine and miconazole on tight junctions, as well as impact of miconazole on dapivirine tissue permeability.
It is clearly and well-written manuscript, and fits under the scope of the Journal. There are just a few minor suggestions to strengthen the manuscript.
Since many abbreviations are used, I suggest the authors to carefully check that they are correctly introduced. For instance, meaning (or full name) for abbreviations „IPM 015“ (line 50) and „PK“ (line 74), please check all others used in the manuscript.
Preparing the list of abbreviations would further increase the clarity of the paper.
pH of the buffer(s) used should be denoted.
The results of the performed investigation demonstrated no influence of dapivirine-miconazole interaction on the transporters, altered tight junction integrity and tissue permeability. It would be appreciated if the authors in some way indicate further step in research of understanding the clinically observed increased systemic exposure of dapivirine with simultaneous administration of miconazole.
Author Response

(The authors gave the same response as above.)
